



# Examining bias in pollen-based quantitative climate reconstructions induced by human impact on vegetation

Wei Ding[1], Qinghai Xu[2], Pavel E. Tarasov[1]

[1]Institute of Geological Sciences, Palaeontology, Free University of Berlin, Berlin, 12249, Germany
[2]Institute of Nihewan Archaeology, Hebei Normal University, Shijiazhuang, 050024, China

*Correspondence to*: Qinghai Xu (xuqinghai@mail.hebtu.edu.cn)

**Abstract.** Human impact is a well-known confounder in pollen-based quantitative climate reconstructions as most terrestrial ecosystems have been artificially affected to varying degrees. In this paper, we use a 'human-induced' pollen dataset (H-set) and a corresponding 'natural' pollen dataset (N-set) to establish pollen-climate calibration sets for temperate eastern China (TEC). The two calibration sets, taking a Weighted Averaging Partial Least Squares (WA-PLS) approach, are used to reconstruct past climate variables from a fossil record, which is located at the margin of the East Asian Summer Monsoon in north-central China and covers the late glacial–Holocene from 14.7 ka BP (thousand years before AD 1950). Ordination results suggest that mean annual precipitation ($P_{ann}$) is the main explanatory variable of both pollen composition and percentage distributions in both datasets. The $P_{ann}$ reconstructions, based on the two calibration sets, demonstrate consistently similar patterns and general trends, suggesting a relatively strong climate impact on the regional vegetation and pollen spectra. However, our results also indicate that human impact may obscure climate signals derived from fossil pollen assemblages. In a test with modern climate and pollen data, the $P_{ann}$ influence on pollen distribution decreases in the H-set while the human influence index (HII) rises. Moreover, the relatively strong human impact reduces woody pollen taxa abundances, particularly in the sub-humid forested areas. Consequently, this shifts their model-inferred $P_{ann}$ optima to the arid-end of the gradient compared to $P_{ann}$ tolerances in the natural dataset, and further produces distinct deviations when the total tree pollen percentages are high in the fossil sequence (i.e. about 40% for the Gonghai area). In summary, the calibration set with human impact used in our experiment can produce a reliable general pattern of past climate, but the human impact on vegetation affects the pollen-climate relationship and biases the pollen-based climate reconstruction.

## 1. Introduction

Pollen analysis was initially developed one hundred years ago for inferring past changes in vegetation and climate (von Post, 1946). Since the 1970s, quantitative reconstructions from biological proxies made a revolutionary change to studies of the past climate (Imbrie and Kipp, 1971; Juggins, 2013). Numerical methods such as the Modern Analogue Technique (MAT; Overpeck et al., 1985), Weighted Averaging Partial Least Squares (WA-PLS; ter Braak and Juggins, 1993), and others (Birks et al., 2010; Juggins and Birks, 2012) are widely used in pollen-based quantitative reconstructions (Guiot, 1990; Markgraf et al., 2002; Seppä et al., 2004; St. Jacques et al., 2008; Tarasov et al., 2011; Xu et al., 2010b). Palaeoclimatology relies on modern pollen-climate relationship studies (Li et al., 2009; Markgraf et al., 2002; Seppä et al., 2004; Shen et al., 2006) as well as pollen-climate data compilations (Prentice et al., 2000; Tarasov et al., 2005; Whitmore et al., 2005; Yu et al., 1998; Zheng et al., 2014). Thousands of modern pollen samples from bioclimatic regions all over the world have been collected and analysed, for example, in the framework of the BIOME6000 Project (Prentice et al., 2000). These pollen data have been used for testing the biome reconstruction method and regional sensitivity (e.g. Tarasov et al., 1998; Yu et al., 1998) and for quantitative climate reconstructions using statistical approaches. The principle of uniformitarianism is implicit in these studies and they require modern organism-environment relationships as calibration models (Birks et al., 2010; Juggins and Birks, 2012).





There are several types of uncertainties in reconstructing palaeoclimate from pollen data using calibration models (Guiot et al., 2009; Marquer et al., 2014; Parnell et al., 2016; Xu et al., 2016b). In China and other regions with long-term human occupation, biomes can be strongly modified rather than natural (Ren and Beug, 2002; Zhang et al., 2010). The question of 'how well modern samples reflect the natural vegetation?' thus needs to be addressed (Xu et al., 2010a). It is most likely, that

modern pollen-climate relationships in such regions are different from what they were in the past. For example, comparing the performance of pre-disturbance (1895–1924) and modern (1961–1990) pollen-climate calibration sets from Minnesota, St. Jacques et al. (2008) found that the pre-settlement model performs better than the modern one in reconstructing past climate. The human impact on the terrestrial vegetation over the past 150 years in the American mid-west is thus apparent in the modern calibration set. Such a distortion in the modern pollen dataset can generate bias in the climate reconstruction for those regions

(Li et al., 2014; St. Jacques et al., 2015; St. Jacques et al., 2008; Tian et al., 2017). Palynologists therefore have to face this challenge in vegetation and climate reconstructions when using pollen data from densely populated regions (Juggins and Birks, 2012; Seppä et al., 2004; Tarasov et al., 1999; Xu et al., 2010a).

In China, rich archaeological evidence suggests that crop domestications may have taken place in the early Holocene or even earlier (Bestel et al., 2014; Lu et al., 2009; Zhao and Piperno, 2000), and enhanced farming practices are reported since 8,000

years ago (Liu et al., 2015; Lu et al., 2009; Zhao, 2011). Early agriculture was usually accompanied by slash-and-burn clearance of forest patches (Ruddiman, 2003), and pollen-inferred anthropogenic impacts on natural vegetation are noted from as early as 6,000 years ago in eastern China (Ren and Beug, 2002; Wang et al., 2010). Due to growing demand for land, construction materials and fuel, disturbance to the natural vegetation disturbances over the last two millennia occurred widely and are commonly detected in the pollen records (Cao et al., 2010; Ni et al., 2014; Xu et al., 2016a; Zhao et al., 2010; Zhao et

al., 2009). Consequently, human impact in both the modern reference datasets and the fossil pollen records needs to be considered when reconstructing past climate from pollen. In China, in contrast to North America, it is not possible to establish a calibration set consisting of 'pre-settlement' pollen and climate data. However, it is still important to estimate what kind of bias may appear in pollen-based quantitative climate reconstructions using Chinese pollen data.

In the past two decades, a number of modern pollen studies have been conducted in China to investigate regional pollen-

vegetation-climate relationships (Herzschuh et al., 2010; Li et al., 2009; Lu et al., 2011; Luo et al., 2009; Shen et al., 2006; Xu et al., 2007; Zhang et al., 2012; Zheng et al., 2008) and human impact on vegetation (Ding et al., 2011; Liu et al., 2006; Pang et al., 2011; Wang et al., 2009; Yang et al., 2012; Zhang et al., 2014; Zhang et al., 2010). At the same time, representative modern reference datasets (Cao et al., 2014; Xu et al., 2010a; Yu et al., 2000; Zheng et al., 2008; Zheng et al., 2014) and fossil pollen datasets (Cao et al., 2013; Ren and Beug, 2002; Sun et al., 1999) have been assembled, which make it possible to

reconstruct the vegetation and climate for individual sites, regions or the whole China (Chen et al., 2015; Ni et al., 2014; Tian et al., 2016; Wang et al., 2014; Xu et al., 2010b). Despite the aim of these studies to use modern surface samples from natural (i.e. likely undisturbed) vegetation communities for establishing their calibration datasets, and to exclude samples representing human-disturbed vegetation communities (Zheng et al., 2014), the presence in the datasets of some samples from eastern China referred to as 'troublesome' due to intense human impact surrounding the vegetation patches (Xu et al., 2010a), suggests that

the problem was not completely resolved. Li et al. (2014) assessed the reference pollen data in the currently available datasets from central eastern China using a human influence index (HII), and concluded that surface samples are biased due to significant human impact on the natural vegetation. Pollen-based climate reconstructions for the late Holocene in this region would thus also be biased.

Obtaining reliable climate reconstructions in regions with a long history of human activities is indeed a big challenge but also

an important scientific task. For example, reconstructing rainfall in temperate eastern China (TEC) is not only necessary for understanding the East Asian summer monsoon (EASM) variations (Chen et al., 2015; Guiot et al., 2008; Wen et al., 2013; Xu et al., 2010b), but also very important when studying human adaptation to climate change, as well as the origins of



agriculture and cultural evolution (Lu et al., 2009; Mu et al., 2015; Tarasov et al., 2006). In this paper, we (1) compiled a 'human-induced' training set with 791 surface pollen spectra and a corresponding 'natural' training set with 806 spectra from TEC, (2) compared the pollen-climate model performances of the two calibration sets, (3) investigated the deviations of the reconstructed results for a fossil record based on each calibration set, and (4) discuss the mechanism of bias caused by human impact.

## 2. Regional setting

Temperate eastern China (TEC: 30–53 °N, 100–135 °E) was chosen as the study area for its ecological sensitivity to climate change (e.g. forest-steppe boundary shifts with EASM variations) and long-term human impact on vegetation (Guiot et al., 2008; Liu et al., 2014; Ren, 2000; Xiao et al., 2004). The region extends from the eastern margin of the Tibetan Plateau (TP) to the Yellow Sea coastline and from the northern catchment of the Yangtze River to the Heilong (Amur) River, covering about one third of China (Fig. 1a). Topographically, it encompasses three distinct levels, showing a decrease in elevation from the TP margin (2000–4000 m), to the Inner Mongolia Plateau, Loess Plateau, Qin Mountains, Taihang Mountains, and Greater Khingan Mountains (1000–2000 m), and to the eastern hilly areas and flood plains (<200–500m). The south-eastern part of the study area is dominated by EASM, while the north-western part is influenced by the Westerlies (Fu et al., 2008). From the coast to inland, mean annual precipitation ($P_{ann}$) varies from 1400 to 35 mm, covering the conventional humid, sub-humid, semi-arid and arid areas, and mean annual temperature ($T_{ann}$) decreases from 18 to -6 °C from south to north (Domrös and Peng, 1988).

Due to the large climatic and topographic gradients, several large-scale natural vegetation regions have been described for the study area (Fig. 1): (I) cold-temperate needleleaf deciduous forest region, (II) temperate mixed needleleaf and deciduous broadleaf forest region, (III) warm temperate deciduous broadleaf forest region, (VI) temperate steppe region, (IVAi) northern subtropical broadleaf evergreen-deciduous forest zone, (VIIBi) temperate semi-shrub and shrub desert zone, and (VIIIAi) subalpine scrub and alpine meadow zone (Editorial Committee of Vegetation Map of China, 2007; Wu et al., 2013). However, many natural ecosystems have been intensely modified by settlements and agricultural land use. For example, in 2015, forest coverage in the supposedly densely forested north-east China was about 41% and only 26% in the warm temperate forest region, according to data from China National Bureau of Statistics (http://data.stats.gov.cn).

## 3. Materials and methods

### 3.1 Surface pollen data

We use pollen data from a number of studies attempting to detect human-induced changes in surface pollen assemblages, including from the Anyang area, central China Plain (Wang et al., 2009), Hexi Corridor and Xinjiang (Ma et al., 2009), warm temperate hilly areas (Ding et al., 2011), Hebei Plain and adjacent mountains area (Pang et al., 2011), south-east China (Yang et al., 2012) and north-east China (Li et al., 2012; Li et al., 2015). Additionally, 70 unpublished spectra from the coastal plain between the Yellow River and the Yangtze River were generated for the purpose of this study. The samples were mostly collected from croplands, abandoned croplands, economic gardens and forests, pasturelands, and roadside scrub and woodlands. The field sampling strategies, laboratory procedures, analytical methods, pollen taxa and other detailed information are described in the aforementioned studies. Additionally, we make use of some reference pollen datasets, partly or entirely covering the study area (Wen et al., 2013; Xu et al., 2010a; Xu et al., 2007; Zheng et al., 2008), which were used to represent 'natural' vegetation communities.

Samples from the different natural vegetation communities were integrated into a 'natural' dataset (N-set), while samples from human-induced vegetation or vegetation obviously disturbed by human activities were integrated into a 'human-induced'



dataset (H-set) (Fig. 1b). We used reference samples from an approximate extent of 31 °N–51 °N and 102 °E–130 °E, with a sufficiently large $P_{ann}$ gradient of 150–1100 mm and $T_{ann}$ gradient of -3–16 °C, in order to cover the greatest possible climate range likely to be encountered in the fossil pollen record (see section 3.3). Pollen percentages were recalculated based on the sum of terrestrial taxa. It should be clearly noted that pollen of cereal-type Poaceae and other distinct cultivated taxa (e.g.

*Brassica*, *Gossypium*, *Sesamum* and *Linum*) identified in the H-set (Ding et al., 2011; Li et al., 2015) were excluded to reduce anthropogenic noise. This strategy is similar to the one excluding aquatic pollen and spores in order to better catch the climatic signal. Finally, 806 spectra and 151 taxa form the N-set, and 791 spectra and 147 taxa form the H-set.

### 3.2 Modern climate and Human Influence Index data

Mean monthly climate averages were derived from the latest available observation data (1981–2010) of 1208 well-distributed
meteorological stations across the study area (Fig. 1a). The original data can be accessed from the China National Meteorological Information Center (http://data.cma.cn). Mean values of annual precipitation ($P_{ann}$) and temperature ($T_{ann}$), and mean temperature of the coldest ($Mt_{co}$) and warmest month ($Mt_{wa}$) were selected as the transfer function variables. These four climate parameters were estimated for each pollen site with the Polation 1.1 software (http://polsystems.rits-palaeo.com). A vertical lapse rate of 0.6 °C/100 m, as suggested for China (Domrös and Peng, 1988), was applied and leave-one-out cross-
validation was used to assess the interpolation accuracy. Correlation coefficients (R) between estimated and observed climatic values of 0.97–0.99 suggest the results are robust.

Sanderson et al. (2002) developed a human influence index (HII) for mapping the human footprint and the last wilderness on Earth. HII values range from 0 to 64 (WCS/CIESIN, 2005) and indicate the degree of human impact on an 1 km$^2$ area (Fig. 1b). The index quantifies human influence on terrestrial ecosystems based on human population density, extent of built-up
area, land transformation, roads, railways, navigable rivers, coastline, and electric power infrastructure. The HII has been recently employed to assess human influence on pollen assemblages in China (Li et al., 2014; Li et al., 2015). In this study, HII values at 1-km$^2$ grids are assigned to modern pollen reference sites using ArcGIS.

### 3.3 Fossil pollen record from Lake Gonghai

Lake Gonghai (38°54′N, 112°14′E, 1860 m above mean sea level) is a small (0.18 km$^2$) hydrologically-closed alpine lake with
water supply mainly from summer precipitation (Chen et al., 2015). It is located on the north-east margin of the Loess Plateau (Fig. 1). The lake lies close to the modern East Asian summer monsoon (EASM) border in the forest-steppe ecotone and experiences sub-humid to semi-arid transitional moisture conditions. The upper 9.42 m of a core, GH09B, from Lake Gonghai was sub-sampled at 1-cm intervals for pollen analysis. Twenty-five (including seven from parallel core GH09C) accelerator mass spectrometry (AMS) $^{14}$C dates of terrestrial plant macrofossils and 35 $^{210}$Pb/$^{137}$Cs dates of the uppermost 0.35-m of the
lake sediment were used to establish a robust age-depth model (Chen et al., 2015; Xu et al., 2016a). We used this well-dated and high-resolution pollen sequence to reconstruct past climate, using both the N-set and H-set of modern pollen as clibaration sets.

### 3.4 Numerical analyses

Relationships between surface pollen spectra and climate variables are assessed by ordination techniques. To stabilise the
variance and optimise the signal-to-noise ratio in the data, pollen taxa which occur in at least 3 samples and contribute ≥ 3% in at least one sample were selected and square-root transformed for further analyses (Prentice, 1980). The length of the first axis in Detrended Correspondence Analysis (DCA; Hill and Gauch, 1980) was used to determine whether Redundancy Analysis (RDA) or Canonical Correspondence Analysis (CCA) should be chosen for the constrained ordination (ter Braak and Prentice, 1988). The ratio of the constrained eigenvalue to the first unconstrained eigenvalue ($\lambda_1/\lambda_2$) for a climate variable is
used to assess its potential to be reconstructed (ter Braak, 1987). A value of $\lambda_1/\lambda_2$ greater than one suggests that the variable is



the main determinant in the dataset; otherwise, the reconstruction of the variable should be conducted with caution (Juggins, 2013). HII was also analysed in the same way to evaluate the human impact on the pollen data.

The WA-PLS approach (ter Braak and Juggins, 1993) has been tested, along with other statistical techniques, for eastern China data and demonstrated to give better results (Cao et al., 2014; Xu et al., 2010a) due to its generally good performance under
non-analogue situations and ability to cope with spatial autocorrelation (Cao et al., 2014; Juggins and Birks, 2012). The optimal number of WA-PLS components was selected using a randomisation t-test (van der Voet, 1994). Low root mean squared error of prediction (RMSEP), low average and maximum biases, a high coefficient of determination ($R^2$) between the predicted and observed climate values, and a rule-of-thumb threshold of 5% (reduction in RMSEP for adding a component) were all considered when selecting a model (Birks, 1998; Birks et al., 2010; Juggins and Birks, 2012).

The significance of the obtained reconstructions was also tested. The proportion of variance in the fossil sequence explained by 999 transfer functions trained with random data was calculated from a constrained ordination (Telford and Birks, 2011). To help understand the bias mechanism of human impact on pollen assemblages, we estimated the weighted average (WA) optima and tolerances (Birks et al., 1990; ter Braak and Looman, 1986) of selected climate variables for major taxa. The five closest modern analogues for each fossil sample were calculated using MAT (Simpson, 2007). The mean HII value at the
analogue location site was used to examine the potential human influence on analogue samples, and further, to evaluate the bias in climate reconstruction for that fossil sample. All numerical analyses were performed using *vegan* version 2.3-5 (Oksanen et al., 2016), *analogue* version 0.17-0 (Simpson, 2007), *rioja* version 0.9-5 (Juggins, 2015) and *palaeoSig* version 1.1-3 (Telford, 2015) in R 3.2.4 environment (R Core Team, 2016).

## 4. Results

### 4.1 Relationship between modern pollen and climate

Ordinations are based on square-root transformed pollen data of 99 taxa in the N-set and 93 taxa in the H-set after noise reduction. DCA showed that the length of the first axis is 2.65 SD (standard deviation units) in the N-set and 2.36 SD in the H-set, suggesting that linear ordination techniques (e.g. RDA) are appropriate to present the distribution of pollen taxa along the climate gradients in our datasets. When using each of the climatic variables as a sole predictor, $P_{ann}$ explains 20.56%
(highest) of the pollen assemblage variance in the N-set, while the thermal variables have much lower explanatory power ($T_{ann}$: 2.83%, $Mt_{co}$: 3.49%, $Mt_{wa}$: 6.35%). For the H-set, $P_{ann}$ explains 6.31%, which is slightly less than $Mt_{wa}$ (6.62%). If we assess the marginal contribution of a variable after partialling out the interaction effect of other variables in an RDA, $P_{ann}$ explains the highest amount of variance in both the N-set (10.56%) and the H-set (5.85%). HII explains more variance in the H-set (2.29%) than in the N-set (1.12%), and has a marginal contribution in both the H-set (0.55%) and the N-set (0.76%). $P_{ann}$ has
the highest $\lambda_1/\lambda_2$ ratio in both the N-set (1.28) and the H-set (0.34); the $\lambda_1/\lambda_2$ ratios for all thermal variables and HII are much less than one (Table 1). Our ordination results suggest $P_{ann}$ is the main determinant of pollen distribution in TEC, and $P_{ann}$ in the N-set is used to establish a standard calibration set. We then use the H-set to establish a contrasting pollen-$P_{ann}$ calibration set to compare the deviation in the reconstructions, and to see the extent of the potential bias induced from human impact on the modern pollen assemblages.

### 4.2 Test of the WA-PLS models

A 2-component WA-PLS model performed best with the lowest RMSEP and highest $R^2$ for the H-set, and a 3-component model for the N-set (Table 2). However, the improvement (1.48% reduction in RMSEP) over the 2-component model was less than the threshold of 5%, and therefore we selected a 2-component WA-PLS model for both datasets. The $R^2$ between predicted $P_{ann}$ values and observed values in the N-set is 0.86 and the RMSEP is 89 mm. Both are better than those for the H-set ($R^2$ =





0.64; RMSEP = 100 mm) (Fig. 2). The percentage of RMSEP to the sampled $P_{ann}$ gradient (940 mm) for the N-set is 9.47% and for the H-set (927 mm) 10.79%. Overestimates at the low-end of the $P_{ann}$ gradient (i.e. for sites from arid areas) and underestimates at the high-end (sites from humid areas), which is an inevitable systematic bias in all WA-based models and referred to as 'edge effects', are larger in the H-set than in the N-set.

### 4.3 Annual precipitation ($P_{ann}$) reconstructions for Lake Gonghai

We applied the pollen-$P_{ann}$ WA-PLS models to the Lake Gonghai fossil record (Chen et al., 2015; Xu et al., 2016a). The proportion of the variance in the fossil data explained by the first PCA axis is 47.79%, and the significance tests suggest that 41.57% variance in the N-set can be explained by $P_{ann}$ ($P = 0.001$) and 23.32% for the H-set ($P = 0.033$) (Fig. 3). The two calibration sets produced very similar reconstructed $P_{ann}$ patterns (including major trends and change-points) but with distinct deviations in their values (i.e. range: -105–95 mm and SD = 46 mm) for most of the time (Fig. 4a). The deviation is calculated as reconstructed N-set $P_{ann}$ value minus corresponding H-set $P_{ann}$ value (Fig. 4b). From the deviation pattern, six zones or time windows (TWs) are demarcated. From 14.7 to 13.1 ka BP (TW1), the deviation decreases gradually from -70 to 20 mm. During 13.1–12 ka BP (TW2), the deviation fluctuates between -30 and 30 mm and the mean value for this period is only 3 mm. The deviation varies from -20 to 100 mm and generally appears to increase in the 12.0–7.3 ka BP interval (TW3). A downward trend can be observed starting from 7.3 ka BP (TW4), and the deviation decreases from a relative stable value of 75 mm to around zero ($\pm$35 mm) during 2.8–1.6 ka BP (TW5). In the most recent period (TW6), the deviation is mostly negative with a mean value of -40 mm. The deviation curve over the last 14.7 ka generally correlates (R = 0.81) with the tree pollen percentage curve (Fig. 4c).

### 4.4 WA optima estimates and analogue measures

The WA optima and tolerances of 15 major pollen taxa in both the N- and H-sets were estimated (Fig. 5). The optima for $P_{ann}$ for most tree taxa (*Picea*, *Pinus*, *Betula*, *Quercus*, *Carpinus*, *Juglans*) and for *Corylus* (representing shrubs and small trees) in the H-set are shifted to drier conditions compared to that in the N-set, while for most herbaceous taxa (Chenopodiaceae, *Polygonum*, Poaceae, *Artemisia*, Ranunculaceae) and the drought-enduring shrub *Nitraria* the optima are shifted towards wetter conditions. *Ulmus* (a commonly cultivated tree) and Cyperaceae (species-rich herbaceous taxon) are exceptions in the arboreal and non-arboreal groups, respectively. The estimated range of tolerance in the H-set is compressed for most taxa in comparison to the N-set, especially for tree taxa, which shrink by about 18–55% (Fig. 5). The mean HII value of the five best modern analogues in the H-set are generally higher than those in the N-set, except for 13.1–12 ka BP and the last 1.6 ka period when they are relatively close; both are low for the mid-Holocene fossil samples (Fig. 4d).

## 5. Discussion

### 5.1 Climatic signals in pollen assemblages obscured by human impact

The pollen record is a complex and nonlinear function of vegetation which in turn is a function of climate (Birks et al., 2010). The big challenge for pollen-based climate reconstructions is that this indirect pollen-climate relationship can be affected by several other (non-climatic) factors, for example, by human activities (Birks and Seppä 2004; Ren, 2000; Xu et al., 2010b). RDA results show that the ability of $P_{ann}$ to explain pollen variance declines a lot in the H-set in comparison to the N-set (Table 1). The statistical performance of WA-PLS for the H-set is poorer (Table 2), suggesting climatic signals in the H-set have been partly obscured by human impact. Due to agricultural land use and human-induced deforestation of the plains and hilly areas (e.g. terraced fields) in the humid and sub-humid regions, tree pollen percentages decrease and herb percentages increase substantially in surface samples, even after excluding distinctly cultivated taxa. It is easy to imagine that herbaceous taxa, such as *Artemisia*, Chenopodiaceae, *Humulus* and weed-type *Poaceae* would expand after forest clearance and this will change the





regional vegetation composition and relative pollen abundances (Ding et al., 2011; Li et al., 2015). It will also alter the pollen-climate relationships for many pollen taxa in the response models (St. Jacques et al., 2008). This alteration can be seen in the comparison between the estimated WA optima and tolerances for major taxa in the two datasets of this study (Fig. 5).

Selected pollen taxa can be separated into two tree and herb groups by contrasting their WA optima in the N- and H-sets (Fig. 5). The inferred optima of most woody taxa in the H-set are shifted towards drier conditions and their tolerances compressed. This means that a fossil pollen assemblage with a high proportion of woody taxa would be assigned a lower $P_{ann}$ value when the H-set is employed in the transfer function. Conversely, $P_{ann}$ values will be overestimated when herbaceous taxa dominate in a fossil sample. This is clearly seen in the $P_{ann}$ curves for Lake Gonghai (Fig. 4a). The reconstructed $P_{ann}$ deviations between the N- and H-sets and the tree pollen percentage curve demonstrate similar trends (Fig. 4) and are statistically correlated (R = 0.81). However, the $P_{ann}$ deviation is not simply determined by the proportion of tree and herb taxa. For example, the deviations of (time-window) TW2 and the later TW5 are both around zero, but the tree pollen comprises 20–30% and 40–50%, respectively, rather than being equal with the percentage of herbs. This is a consequence of the vegetation composition and species characteristics.

In short, human impact obscures the climatic signals in pollen spectra by distorting the response relationship between pollen abundance and climate (Birks et al., 2010; Seppä et al., 2004), thus influencing the assumed climatic optima and tolerances of pollen taxa in the model (Fig. 5). When such a human-influenced calibration set is applied to a fossil record, which represents mostly natural vegetation, a more or less serious bias in the reconstructed past climate should be expected (St. Jacques et al., 2008; Xu et al., 2010a). Using the H-set in this study, a significant bias towards drier conditions (relative to the N-set) is found for Lake Gonghai during the early and middle Holocene when tree pollen contributed more than about 40% to the total pollen sum. Conversely, we note a bias towards a wetter climate reconstruction for the late glacial (TW1) and last 1600 years (TW6). The sites comprising the two sets, N and H, are not perfectly even distributed which may also influence the optima estimates (Fig. 1b). However, the model inferred group-optima change pattern is statistically and ecologically reliable (Fig. 5). We consider it likely that a similar effect will occur in the pollen-based climate reconstructions for the whole TEC region, where vegetation pattern has been largely shaped by EASM-induced rainfall.

### 5.2 Human Influence Index as an assessment tool

Modern HII captures basic characteristics of human influence on ecosystems and allows a quantitative evaluation of human impact on the land surface (Sanderson et al., 2002). Li et al. (2014) innovatively employed HII to establish a calibration set with pollen data and applied it to a 6200-year fossil record from Lake Tianchi in central China (Zhao et al., 2010). The pollen-HII calibration model ($R^2 = 0.47$) was based on 185 modern samples from central eastern China (belong to the warm temperate forest region), and the variance in the training set explained by HII (6.79%) is comparable to $P_{ann}$ (7.78%) and $T_{ann}$ (6.00%) (Li et al., 2014). A further investigation based on 189 surface pollen samples from northern China (involving both human-induced and natural samples from vegetation regions II, III and VI) provided a higher correlation ($R^2 = 0.69$) between pollen and HII values in a WA-PLS model (Li et al., 2015). This good statistical performance gives us more confidence in assessing human influence on vegetation using the HII, although we are aware of some difficulties in applying a quantitative pollen-HII calibration model to the fossil data.

A good correlation of the HII data with cereal-type Poaceae pollen in northern China (Li et al., 2015) suggests that the HII can be seen as a surrogate of indicator pollen taxa for human activities. However, cereal-type Poaceae pollen generally has a very low abundance in a fossil sequence. For example, the cereal-type Poaceae in a 'natural' profile close to the archaeological sites from Anyang – the centre of agricultural and societal development during late Shang Dynasty – comprises only around 2% of the total pollen during the last 3,400 years (Cao et al., 2010). In the sequence from Lake Gonghai used in the current study it contributes about 2–4% during the last two millennia (Xu et al., 2016a). Therefore, HII explains only 1.12% of the variance in





the N-set and 2.29% in the H-set after removing the cereal-type Poaceae and other distinct cultivars (Table 1). In addition to cereal-type Poaceae, taxa such as *Artemisia*, Chenopodiaceae, and weed Poaceae, which could be dominant in both steppe areas and sub-humid areas after forest clearance (Ding et al., 2011; Li et al., 2008; Liu et al., 2006), may challenge the interpretations. Although human impact can be detected using additional information, including charcoal and archaeological

data (Zhao et al., 2010), compositional change of these taxa during the late Holocene due to human activities are hard to distinguish from those caused by a progressively drier EASM climate, especially in fossil pollen records from the forest-steppe ecotone in TEC.

Reconstructing human influence quantitatively from fossil pollen data with a direct pollen-HII calibration set might not be an easy task in most cases (Li et al., 2014), but we can still use HII as an assessment tool in a broad-spectrum way. The

reconstructed climate of a certain fossil sample is mostly determined by its closest modern analogues even though different approaches may have been used for the reconstruction (e.g. WA-PLS) (Birks et al., 1990). By examining the mean HII values at sites of the best modern analogues, we can evaluate the bias in the climate reconstruction of the corresponding fossil sample. A high analogue-HII value indicates greater potential bias in the reference samples. As shown in Fig. 4d, analogue-HII values in the H-set are usually higher than in the N-set, suggesting a higher bias in the $P_{ann}$ reconstruction. The mean analogue-HII

values in the N-set fluctuate around 15 during the mid-Holocene indicating that the climate reconstruction for this interval has the lowest human-induced bias as a HII value of 15 is the mean background value for modern 'natural' vegetation patches in TEC (Li et al., 2015; Sanderson et al., 2002). Similar to the HII trend in Lake Tianchi (Li et al., 2014), analogue-HII values for Lake Gonghai start to rise around 2.8–2.9 ka BP, which conforms to the scenario of agricultural advancements and population growth in Bronze Age China during the Western Zhou period (1045–771 BC) (Li, 2006). Relatively higher

analogue-HII values during the late glacial and early Holocene suggest that modern analogues for this period in the current reference dataset have experienced more human influence. Together with the common problem of no analogues for this period (Jackson and Williams, 2004), climate reconstructions for this interval in TEC should be considered more carefully.

### 5.3 Implications for Holocene climate reconstructions

Agriculture became the dominant subsistence strategy in today's (potentially) warm temperate forest region (III, including the

central China Plains) and northern subtropical mixed forest zone (IVAi, including the Yangtze Plains) from about 6.5–5 ka BP (Crawford, 2011; Zhao, 2010). Potential human disturbance to the vegetation in eastern China since 6 ka BP has been inferred from many pollen studies (Ren and Beug, 2002; Wang et al., 2010), not to mention historical times (Cao et al., 2010; Zhao et al., 2010). Our analogue-HII assessments indicate that the bias in the climate reconstruction induced from human impact via reference pollen samples or via changes in the fossil pollen assemblages (particularly during historical time) (Li et al., 2014)

are real. This raises the question of whether the Holocene climate could be quantitatively reconstructed using pollen data from eastern China. The answer is not a simple 'yes' or 'no' (Ren and Beug, 2002), although we keep an optimistic view based on the comparison results presented in this study.

Reconstructed $P_{ann}$ for the uppermost 17 samples from the Gonghai record, representing AD 1950–2008, is 428 mm (SD = 26 mm) with N-set and 439 mm (SD = 25 mm) with the H-set. The modern mean $P_{ann}$ around Lake Gonghai area is 445 mm

according to instrumental data between 1959–2011 from Ningwu Station, located 11 km north of the lake (Chen et al., 2015). Both calibration sets provide reliable (more or less similar) reconstructions for recent decades when compared with historical measurements, although deviations exist for most of the time since the last deglaciation (Fig. 4). By examining the deviation range between the two datasets, and bearing in mind that surface reference samples in the H-set are more or less strongly affected by human activities, we can assume that the $P_{ann}$ reconstruction based on the N-set should be closer to reality. Holocene

pollen-climate relationships in China are relatively stable (Tian et al., 2017). If we select appropriate sites, such as Lake Tianchi and Gonghai, both of which are small closed alpine lakes with very limited human impact before 3 ka BP (Institute of





Archaeology CASS, 2003, 2004), we can still hope to get a reliable $P_{ann}$ reconstruction prior to the late Holocene. Our Holocene $P_{ann}$ reconstruction using the N-set can further the discussion of cultural evolution and the origin of dry-land agriculture in the study region. For example, a remarkable $P_{ann}$ increase from 480 to 570 mm along the present-day EASM margin during 8.6–7.8 ka BP could have promoted the development of millet agriculture (Liu et al., 2012; Zhao, 2011). It also supports the hypothesis that in early Neolithic sites broomcorn millet (low $P_{ann}$ requirement: 350–450 mm) is more abundant than foxtail millet (optima: 450–550 mm) as broomcorn millet is better adapted to drought conditions (Lu et al., 2009).

Regarding the potential human-induced bias in fossil records, the challenge for pollen-based quantitative climate reconstructions is more from the lack of natural surface samples in the regions with intensive agricultural activities. In eastern China, a calibration set only including pollen samples from lake surface sediments with low human disturbance is still not available (Liu et al., 2013), and surface samples are mostly collected from mountains and steppe areas (Xu et al., 2010b). This means that our current modern pollen datasets (Cao et al., 2014; Zheng et al., 2014) still contain relatively few samples from central eastern China. Collecting extra samples from 'natural' vegetation in the mountain areas, such as Luzhong, Qin, Dabie, and Qian mountains would help to improve the pollen-based climate reconstructions for the region. The Qin mountains, for example, have a large and well-forested range (ca. 57,000 km$^2$), but are represented by relatively few pollen samples (Fig. 1b). In addition, there are still hundreds of small forests (patches) in the hilly areas, around the lakes, and within natural parks in eastern China, which deserve the attention of palynologists.

## 6. Conclusions

This paper attempts to assess the extent of bias induced from human impact in pollen-based quantitative climate reconstructions. Numerical analyses suggest that $P_{ann}$ is the main explanatory variable for pollen distribution in temperate eastern China, even in the pollen dataset with intense human impact. Model-inferred $P_{ann}$ optima of most major woody pollen taxa in the human-induced dataset shift to the arid-end of the gradient, and resulting in the underestimation of $P_{ann}$ when the percentages of tree pollen is high in the fossil record. In the context of long-term human impact on vegetation in the study region, a bias in pollen-based climate reconstructions is inevitable. However, our study demonstrates how this bias is manifest and how a more reliable $P_{ann}$ reconstruction can be inferred from the fossil pollen record. Reconstructed $P_{ann}$ using the 'natural' dataset in this study reliably portrays the Holocene monsoon rainfall variations in northern China and supports a valid interpretation of the dry-land agriculture origin in the region. Our research also indicates that climate reconstructions should be conducted with caution, particular for the last one or two millennia when population pressure is high and land use is intensive. Other sources of evidence, including archaeological or historical data are helpful (and absolutely necessary) for a more accurate interpretation of results.

## Acknowledgements

We are grateful to Dr. Chunhai Li for his pollen analyses of samples from Jiangsu coastal plain and Cathy Jenks for her linguistic help. This work was supported by the key programs of National Natural Science Foundation of China (40730103 and 41630753). The doctoral research of W. Ding at Freie Universität Berlin in the working group of P. Tarasov was funded by the China Scholarship Council (2011813072).

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





**Table 1.** Summary statistics for redundancy analysis (RDA) with pollen species and climate variables (annual precipitation $P_{ann}$, mean annual temperature $T_{ann}$, mean temperature of the coldest month $Mt_{co}$, mean temperature of the warmest month $Mt_{wa}$) and the human influence index (HII). Sole expl (%) is the pollen variance explained by variables as a sole predictor, Marg expl (%) is the marginal contribution of this variable in the model with all other variables. All $P$-values are 0.001 (based on 999 unrestricted Monte Carlo permutations).

| Variables | Natural dataset | | | Human-induced dataset | | |
|---|---|---|---|---|---|---|
| | $\lambda_1/\lambda_2$ | Sole expl (%) | Marg expl (%) | $\lambda_1/\lambda_2$ | Sole expl (%) | Marg expl (%) |
| **$P_{ann}$** | **1.28** | **20.14** | **10.56** | **0.34** | **6.31** | **5.85** |
| $T_{ann}$ | 0.08 | 2.83 | 1.20 | 0.26 | 5.57 | 1.55 |
| $Mt_{co}$ | 0.11 | 3.49 | 1.23 | 0.18 | 3.90 | 1.43 |
| $Mt_{wa}$ | 0.21 | 6.35 | 1.37 | 0.31 | 6.62 | 1.44 |
| HII | 0.03 | 1.12 | 0.76 | 0.10 | 2.29 | 0.55 |

**Table 2.** Summary performance statistics of the first three components of the weighted averaging partial least squares regression (WA-PLS) for annual precipitation ($P_{ann}$) based on leave-one-out cross-validation for the natural (N-) set and human-induced (H-) set. Coefficient of determination between predicted and observed $P_{ann}$ ($R^2$), root mean squared error of prediction (RMSEP) (mm), average bias (Ave bias) and maximum bias (Max bias), RMSEP change in percentage (%Change), and $p$-value are given. The selected models are shown in bold.

| Datasets | Model | $R^2$ | RMSEP | Ave. bias | Max. bias | %Change | $p$-value |
|---|---|---|---|---|---|---|---|
| N-set | WA-PLS Component 1 | 0.83 | 96.02 | -0.34 | 82.10 | - | - |
| | **WA-PLS Component 2** | **0.86** | **89.24** | **1.10** | **72.98** | **-7.06** | **0.001** |
| | WA-PLS Component 3 | 0.86 | 87.92 | -0.25 | 64.53 | -1.48 | 0.098 |
| H-set | WA-PLS Component 1 | 0.58 | 108.32 | -1.39 | 218.50 | - | - |
| | **WA-PLS Component 2** | **0.64** | **100.48** | **1.02** | **183.06** | **-7.23** | **0.001** |
| | WA-PLS Component 3 | 0.64 | 100.84 | 0.99 | 177.97 | 0.36 | 0.614 |



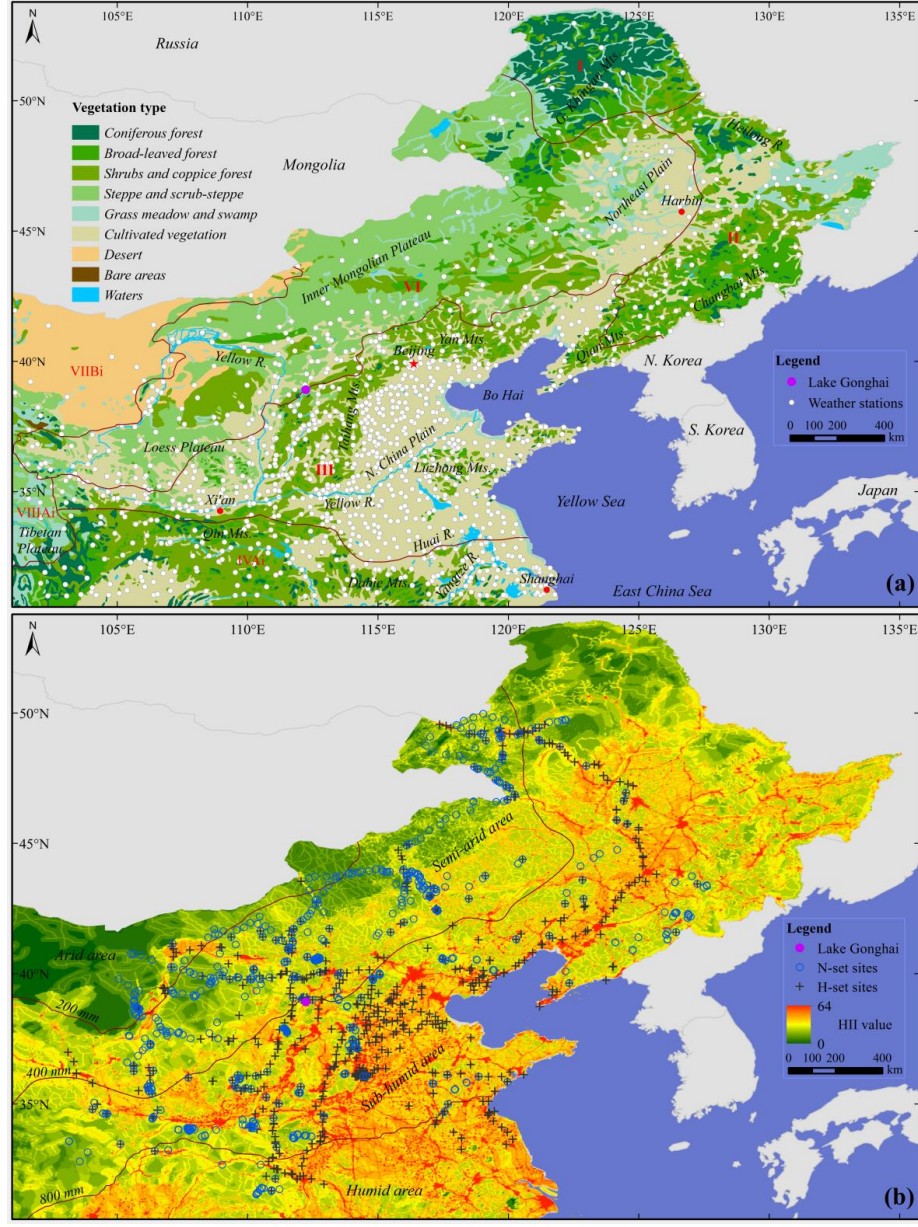

**Figure 1.** Maps of the study region showing the distributions of (a) vegetation regions and types, and the 1208 meteorological stations, and (b) HII values with surface pollen sampling sites from N-set (circles) and H-set (crosses). Selected metropolises, mountains (Mts.), rivers (R.), isohyets of 200, 400 and 800 mm (for conventional climate classification), and Lake Gonghai are marked.



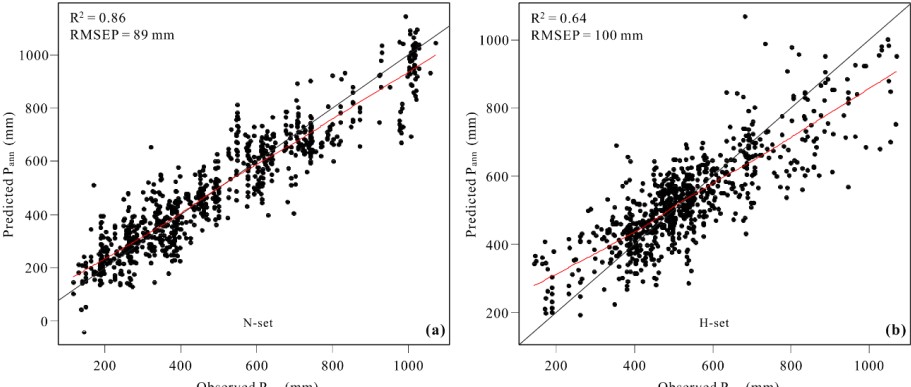

**Figure 2.** Scatter plots of pollen-based predicted annual precipitation ($P_{ann}$) and observed $P_{ann}$ using 2-component weighted averaging partial least squares (WA-PLS) models for (a) the natural set and (b) the human-induced set.

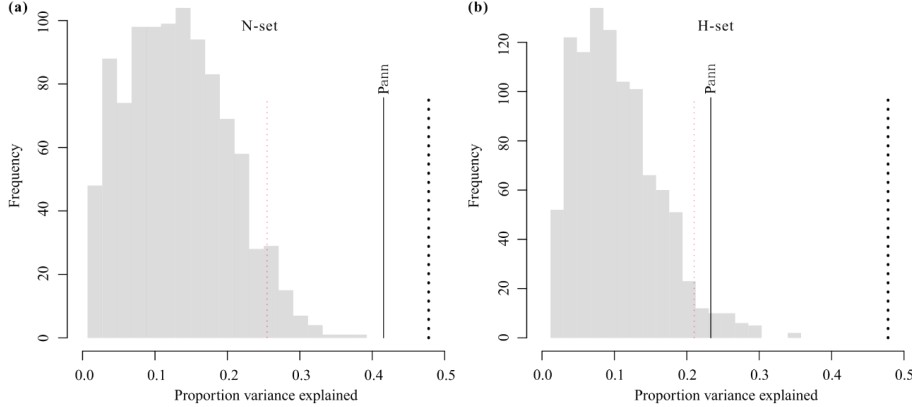

**Figure 3.** Proportion of variance (solid lines) in Gonghai Lake fossil pollen data explained by annual precipitation ($P_{ann}$) transfer functions with (a) the natural (N-) set and (b) the human-induced (H-) set. The thick black dotted lines indicate the

10   proportion of variance explained by the first axis of a principal components analysis (PCA) and the fine red dotted lines indicate the 0.05 significance level. Histograms show the amount of variance explained by 999 transfer functions with random data.





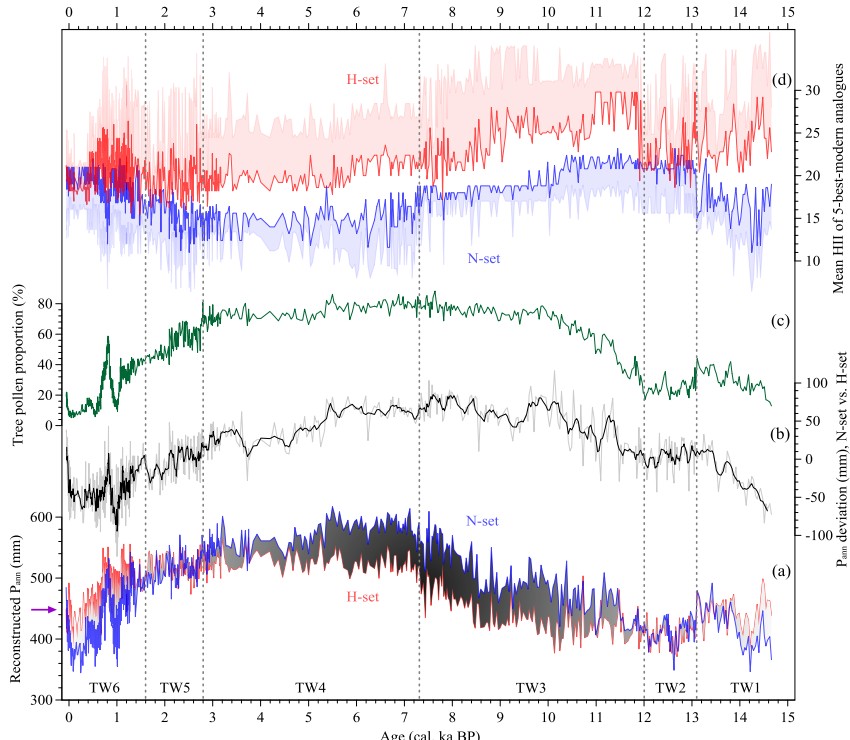

**Figure 4.** Fossil pollen record of Lake Gonghai: (a) reconstructed annual precipitation using a natural (N-) set (blue) and a human-induced (H-) set (red) with a greyscale fill to show the deviations. The modern instrumental value is marked on the scale axis for comparison (purple arrow). (b) Deviations in the reconstructed precipitation (grey) between the N-set and H-set based transfer functions with a 5-point moving average smoother (black). (c) The proportion of tree pollen taxa (%), and (d) the mean HII value of the five best analogues in the N-set (blue, with lower side standard deviation) and the H-set (red, with higher side standard deviation) for fossil samples. Six time windows (TWs), delineated according to the deviation pattern between the two reconstructions, are separated by grey dashed lines.





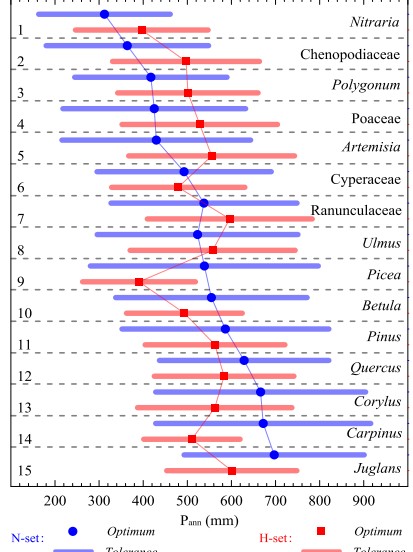

**Figure 5.** Caterpillar plot of weighted average (WA) optima and tolerances for 15 major pollen taxa in response to annual precipitation ($P_{ann}$). The taxa are arranged by optima values and taxa groups. Human impact generally shifts the inferred optima of woody taxa and herb taxa in opposite directions and compresses the tolerances for most taxa in the study area.