# Peer review of "Examining bias in pollen-based quantitative climate reconstructions induced by human impact on vegetation in China"

_Climate of the Past, 2017_

## Referee Comment (RC2) · Anonymous Referee #2 · 29 Jun 2017

I am grateful for the opportunity to review the article entitled "Examining bias in pollen-based quantitative climate reconstructions induced by human impact on vegetation" by Wei Ding, Qinghai Xu, Pavel Tarasov, for *Climate of the Past.*

I think this paper is well written, very clear and concise, and sufficiently interdisciplinary in content to be published in *Climate of the Past*. This study focuses on two modern pollen data calibration sets, one considered as "natural", one with high human impact. These 2 modern pollen datsets are tested to reconstruct the annual precipitation changes during the late glacial and the Holocene in north-central China, at the margin of the East Asian Summer; they used the classical Weighted Averaging Partial Least Squares (WA-PLS) approach to reconstruct quantitatively the precipitation.

This study is very interesting given that the problem of human impact on pollen data is a key problem in climate reconstructions. I think that the paper of Ding et al. presents interesting findings in terms of results; in particular they support the consensus that climate reconstruction based on pollen data must be taken with caution for recent periods (Bronze Age to 0 ka).

So, I recommend the publication with minor changes, which are listed below.

**Main points**

- I think that test two differents datasets -, one considered as "natural", one with high human impact - is really interesting. But in fact, transfer functions to reconstruct past climate are based on the principe that vegetation and climate are in equilibrium-which is not always true, I know-. We keep only pollen samples which have been collected in « natural » ecosystems, we exclude samples which have been collected in areas with high human impact. One another test interesting to do is to test the reconstruction with an unique dataset (natural and non natural) to test if the results will be different or not because we will never use a dataset with human impact.

- what is the definition of « natural vegetation » given the changes made by human societies on their environement , particularly during the late Holocene? How do you define natural vegetation ? is it potential vegetation?

- you study takes only into account the bias linked to modern pollen datasets ; what about the potential biases on fossil data linked to past human impact (fires …); could you more discuss this point?

- I have not seen a description of the fossil record used. Even if it has been already published, it's an important point. Could you add a simplified pollen diagram for the fossil pollen sequence? Could you give details on pollen assemblages? Or describe it very briefly? Time period covered?

- Did you calculate error bars? I have not seen them in the figure. Errors bars are needed before publication to discuss more in depth the possible bias with the H set. And to see if the differences between the 2 climate reconstructions are significant or not.

**Minor points**
1. Title: could you add "in China" in the title?
2. The abstract is well written and informative. May be you can precise in the abstract that the biais in the pollen –based reconstruction is important only during recent periods (not entire lateglacial and Holocene)
3. Introduction
   - p.1, line 26: the ref Von Post, (1916) is more appropriate

- p 1, line 30: I don't agree with the sentence: Palaeoclimatology relies on modern pollen-climate relationship studies; Palaeoclimatology is based on pollen but also on various proxies (speleothems, lakes, tree-rings…) not only pollen; please correct.
- p.1, line 32-33: the ref Bartlein et al., 2011 is missing
- p.1, line 36: what is "the principle of uniformitarianism"?
- p.2, line 5: "in such regions": which ones?

4. Material and methods
   - p.3, line 28: how many pollen data do you use from each region and ecosystem types?
   - p.3, line 37: how do you define the "natural vegetation communities? Do you have pollen traps and vegetation surveys?
   - p.4, line 6: is the pollen sum enough high after exclusion of anthropic taxa, for example in cultivated lands? line 7: it will be informative to precise how many pollen spectra you have per ecosystem (both dataset).
   - p.4, lines 11-12: do you also have calculated GDD5 and the moisture index (prentice et al…)?
   - p.4, line 14: how do you correct the values for the precipitation parameter?
   - p.4, lines 18: more details on the calculation of HII are needed: HII is based only on modern human impact, it doesn't take into account the human impact during recent periods (from 2800 to 0).
   - p. 4 part 3.3 see main point 4
   - p.5, line 3: which statistical techniques have been tested? Could you write 1 sentence on the concept of the WAPLS?
   - p.5, line 14: I don't understand, you also use MAT? Why? Please explain more and give more details on the methods

5. Results
   - p.5, line 21: "the differences between 151(147) and 99(93) taxa in the Natural (human) set are only explained by the exclusion of rare taxa?
   - p.5, line 38: I don't understand why you don't keep the 3 component model (even given the threshold of 5%).

6. Discussion
   - P 5, line 31 ….. is a function of climate : yes but also other factors play a role
   - p. 5, line 18: a significant bias: do you mean statistically significant? it has been tested?
   - p.6, line 16: how is calculated the " mean analogue-HII values in the N-set"?
   - p. 6 line 32: "This raises the question of whether the Holocene climate could be quantitatively reconstructed using pollen data from eastern China. The answer is not a simple 'yes' or 'no'"; I think that the question is "when" and "where" given that your reconstruction before 2.8 ka can be considered as robust and climate driven

7. Tables and figures
   - Fig 1: a, b, the legend is not clear (the right one); the site is not easy to find on the fig: please correct

---

## Author Response (AR1)

**Response to the reviewers**

We would like to thank the reviewers for their valuable comments which helped us to improve this manuscript. We have responded to all the comments and questions, answering them one by one. Our responses are in blue script placed after the reviewers' comments. Words that are highlighted in red are to show the modifications done in the revised version.

**Reviewer #1**

This is a very careful and thorough study that uses different type of pollen data-sets to establish the climate-pollen transfer functions. By selecting natural and human-impact pollen spectrum, the bias effect from human impact on the climatic reconstructions was clearly illustrated. The workload is extraordinary (synthesis on a 1600 pollen record) and the methodology is also robust. I thus highly recommended this paper and I trust it will attract wide interest from paleoclimatologists and paleontologists.

I have only one suggestion on this manuscript. It is essential to reveal the relationship between modern pollen and climate, i.e. to illustrate how important the specific climatic variable (annual precipitation in this case) in explaining the pollen communities. This ms has given detailed numbers (in Table 1) but I think an ordination diagram (bi-plot, environmental variables vs pollen taxa, for both natural and human-impacted dataset) illustrating the importance, significance and the interactions among environmental variables will be preferable. That will also clear show the difference in pollen communities between natural and human-impact scenarios.

*We have added a suggested graph to show the pollen-climate and HII relationships. This newly created figure is marked as Figure 3 (see below).*

*There are 99 pollen taxa in the N-set and 93 in the H-set even after excluding some taxa for noise reduction, and it is becoming unreadable when all these information is presented in a plot. Hence, we have selected 15 major pollen taxa which also mainly identified in GH09B fossil sequence to make this biplot. It is clear to show the difference of pollen taxa-environmental variables relationships between two data sets, and it also helpful to understand the model-inferred optima of tree and herb taxa altered by human impact discussed in the text.*

*We also add a brief description in section 4.1.*

*"To better illustrate the modern pollen-climate relationships and their difference between natural and human-impact scenarios, 15 major pollen taxa, which also identified in GH09B fossil sequence (Fig. 2), were selected to reveal the relationship between modern pollen and climate (Fig. 3). It seems that general pattern of tree and shrub-herb group separation is maintained, but relationship of some pollen taxa (e.g. Picea, Betula, Poaceae and Chenopodiaceae) with climatic variables (e.g. $P_{ann}$) is altered by human influence. Greater ordination difference of Poaceae and Chenopodiaceae in two sets indicates that these two taxa are more sensitive to human impact."*

**Reviewer #2**

I am grateful for the opportunity to review the article entitled "Examining bias in pollen-based quantitative climate reconstructions induced by human impact on vegetation" by Wei Ding, Qinghai Xu, Pavel Tarasov, for Climate of the Past.

I think this paper is well written, very clear and concise, and sufficiently interdisciplinary in content to be published in Climate of the Past. This study focuses on two modern pollen data calibration sets, one considered as "natural", one with high human impact. These 2 modern pollen datasets are tested to reconstruct the annual precipitation changes during the late glacial and the Holocene in north-central China, at the margin of the East Asian Summer; they used the classical Weighted Averaging Partial Least Squares (WA-PLS) approach to reconstruct quantitatively the precipitation.

This study is very interesting given that the problem of human impact on pollen data is a key problem in climate reconstructions. I think that the paper of Ding et al. presents interesting findings in terms of results; in particular they support the consensus that climate reconstruction based on pollen data must be taken with caution for recent periods (Bronze Age to 0 ka).

So, I recommend the publication with minor changes, which are listed below.

Main points

- I think that test two different datasets -, one considered as "natural", one with high human impact - is really interesting. But in fact, transfer functions to reconstruct past climate are based on the principle that vegetation and climate are in equilibrium-which is not always true, I know-. We keep only pollen samples which have been collected in « natural » ecosystems, we exclude samples which have been collected in areas with high human impact. One another test interesting to do is to test the reconstruction with an unique dataset (natural and non natural) to test if the results will be different or not because we will never use a dataset with human impact.

*Indeed, scientists normally avoid collecting reference samples from human-induced vegetation for establishing their pollen-climate calibration set. The difference between reconstructed results using all samples and 'good' samples is often tiny as only a few samples with obvious human impact are excluded, and the model mechanism is not clear, although the statistical performance is better for using 'good' samples (Xu et al., 2010). Based on a large size data set using human- impacted pollen samples complied from our previous works, we have a chance to make a control-group to test the performance of 'nature' samples. The reconstructed results (value and statistical performance) using all samples (natural and non-natural) occur between the results using the natural set and the human-induced set. For clearly showing the model-inferred optimum altered by human impact and the largest potential bias, we chose to show the results using the natural set and human-induced set though the latter one is not commonly used in a reconstruction work.*

- what is the definition of « natural vegetation » given the changes made by human societies on their

environment, particularly during the late Holocene? How do you define natural vegetation? is it potential vegetation?

*For most parts in the temperate eastern China, the human influences are more intense than before, hence, it is impossible to establish a surface pollen data set with samples from completely natural vegetation, no matter since when human societies had started to change the vegetation. Here, 'nature vegetation' was more defined in practice instead of in theory. We classify the samples to nature group in field work according to one or more following criteria: collecting samples (1) in natural reserve, (2) in areas where vegetation composition is coincident with its potential scenario, (3) from vegetation community where is hard for human access and without obvious synarthropic plants, (4) away from settlements, roads, paths, farmlands, pasture lands, and other human activities as far as possible.*

- you study takes only into account the bias linked to modern pollen datasets; what about the potential biases on fossil data linked to past human impact (fires …); could you more discuss this point?

*Examining the biases introduced from past human impact is one task of our future works. In this study, we have shown the potential biases during the late Holocene, especially since the Bronze Age. Archeological and historical documents, and even cereal-type pollen evidence are indeed existed, but we have difficulty in quantitatively separating human impact from natural change. Based on the results with our contrasting data sets, it seems that we still can extract plausible climate information (at least for the tendency) from fossil pollen data, nevertheless, the pollen-based climate reconstructions in temperate eastern China should be conducted with more caution and a possible bias range should be kept in mind for further discussions.*

- I have not seen a description of the fossil record used. Even if it has been already published, it's an important point. Could you add a simplified pollen diagram for the fossil pollen sequence? Could you give details on pollen assemblages? Or describe it very briefly? Time period covered?

*We have added the fossil pollen diagram and brief introduction in both text and figure cutline.*

*In secetion 3.3, we add a paragraph:* "Vegetation succession in the area around Lake Gonghai has experienced five major stages during the last 14.7 ka (Fig. 2). Open forests and upland meadows dominated during the last deglaciation (14.7–11.1 ka) and abrupt strengthening of *Artemisia*-dominated mountain steppe association occurred during 13.1–12.0 ka BP. In the early Holocene (11.1–9.6 ka), *Betula*, *Carpinus*, *Ostryopsis*, and *Ulmus* as pioneer tree species spreaded in the landscape. Later (9.6–7.3 ka) mixed forest dominated by *Picea* and *Betula* started to play a greater role. Temperate deciduous trees (e.g. *Quercus*) widely expanded and the mixed broadleaf-conifer forest grew around the lake during the Holocene climatic optimum (7.3–5.0 ka BP). The break up of the climax community started from 5.0 ka BP with the increasing of *Pinus* and the decreasing of *Quercus* and *Betula* percentages. Major increase in herbaceous pollen percentages occureed since 2.8 ka BP, especially during the last 1.6 ka, when *Humulus* (including *Urtica*), *Fagopyrum*, and cereal Poaceae pollen types related to human activities

became most prominent in the diagram (Fig. 2; Xu et al., 2016). Vegetation dynamic inferred from GH09B pollen sequence is a valuable source of environmental information for the EASM margin (Chen et al., 2015). In the current study, we selected this well-dated and high-resolution fossil record to reconstruct past climate, using both the N-set and H-set of modern pollen as calibration sets."

*The new figure was marked as Figure 2 (see below).*

- Did you calculate error bars? I have not seen them in the figure. Errors bars are needed before publication to discuss more in depth the possible bias with the H set. And to see if the differences between the 2 climate reconstructions are significant or not.

*We have added the error bars (standard error of prediction) for the reconstructed $P_{ann}$ values in Figure 6 (as Figure 4 in the previous version of the manuscript, see below).*

Minor points

1. Title: could you add "in China" in the title?

*Added.*

2. The abstract is well written and informative. Maybe you can precise in the abstract that the bias in the pollen-based reconstruction is important only during recent periods (not entire lateglacial and Holocene)

*We added this conclusive information in the end of the abstract.*

"The extent of human-induced bias may be rather small for the entire lateglacial and early Holocene interval when we use a reference set called 'natural'. Nevertheless, this potential bias should be kept in mind when conducting quantitative reconstructions, especially for recent two or three millennia."

3. Introduction

- p.1, line 26: the ref Von Post, (1916) is more appropriate

*Changed.*

- p 1, line 30: I don't agree with the sentence: Palaeoclimatology relies on modern pollen-climate relationship studies; Palaeoclimatology is based on pollen but also on various proxies (speleothems, lakes, tree-rings…) not only pollen; please correct.

*Corrected. We have rewritten this sentence as* "Pollen-based palaeoclimate reconstructions relay on modern pollen-climate relationship studies".

- p.1, line 32-33: the ref Bartlein et al., 2011 is missing

*Added.*

- p.1, line 36: what is" the principle of uniformitarianism"?

*We have rewritten this phrase as: "A methodological assumption that the ecological response of species*

does not change during the Quaternary based on Lyell's uniformitarianism (Scott, 1963)*".

- p.2, line 5: "in such regions": which ones?

*This phrase "in such regions" here means "In China and other regions with long-term human occupation" hereinbefore, and we have rewritten the sentence as: It is most likely, that modern pollen-climate relationships* in such regions with long-lasting human impact *are different from what they were in the past.*

4. Material and methods

- p.3, line 28: how many pollen data do you use from each region and ecosystem types?

*We have added this information in the text.*

"…including 43 spectra from Anyang area, central China Plain (Wang et al., 2009), 12 from eastern Hexi Corridor (Ma et al., 2009), 78 from warm temperate hilly areas (Ding et al., 2011), 88 from Hebei Plain and adjacent mountains area (Pang et al., 2011), 13 from south-east China (Yang et al., 2012), and 105 spectra from north-east China (Li et al., 2012; Li et al., 2015)."

- p.3, line 37: how do you define the "natural vegetation communities? Do you have pollen traps and vegetation surveys?

*We defined the vegetation as "natural" in some sites based on trap samples but mostly relied on vegetation surveys (e.g. Li et al., 2011). As mentioned above in our answer to main point 2, "natural" here was used to refer these vegetation communities which keep their potential characteristics in structure and composition, and without obvious human disturbance.*

- p.4, line 6: is the pollen sum enough high after exclusion of anthropic taxa, for example in cultivated lands? line 7: it will be informative to precise how many pollen spectra you have per ecosystem (both dataset).

*The minimum number of pollen grains to count is 400 in our works, and 500~600 pollen grains were counted for most samples. The mean percentages of cultivated pollen taxa for different type samples are 5~25%, and only up to 60% in some farmland samples. After excluding the human-indicator taxa, most samples still have enough grains (300~500) for further calculations. However, samples with very high percentage of cereal-type pollen were not used in our study.*

*We have added the samples-ecosystem information in the text: "The N-set includes 11 samples from vegetation region I, 61 from region II, 300 from region III, 351 from region VI, 25 from region IVAi, 51 from region VIIBi, and 7 from region VIIIAi, while 14 samples from region I, 11 from region II, 433 from region III, 292 from region VI, and 41 from region IVAi appear in the H-set."*

- p.4, lines 11-12: do you also have calculated GDD5 and the moisture index (prentice et al…)?

*Yes, we had tested the GDD5 and a moisture index (the ratio of evapotranspiration to potential evapotranspiration, AET/PET, as effective moisture, water availability or α index in different literatures)*

*which both are important bioclimatic variables for vegetation distribution. However, the GDD5 is not a determinant (control) variable in the study area and its explanatory power is not better than thermal variables used in the study. For effect moisture, MODIS-based AET/PET data from (Zhao and Running, 2010) was used. It is highly correlated to $P_{ann}$ because our study area mainly including sub-humid and semi-arid regions where $P_{ann}$ is a determinant variable. Therefore, we did not use these two variables in our study.*

- p.4, line 14: how do you correct the values for the precipitation parameter?

*We did not correct the precipitation values with any calibration. The precipitation lapse rate is still not available in our study area. There are some results from sites studies in western China (e.g. the precipitation value increases with the elevation in Tianshan area and the multi-year average rate are 6~11 mm per 100m for different slopes), however, we are not sure it can be used in monsoon region or not. Our pollen-climate data set is the first time to use new available and high-resolution observation data from 1208 meteorological stations, which are well-distributed in the study area and cove the elevation range of 0~4000 m. The interpolation of precipitation values for pollen sites is credible.*

- p.4, lines 18: more details on the calculation of HII are needed: HII is based only on modern human impact, it doesn't take into account the human impact during recent periods (from 2800 to 0).

*We have rewritten the paragraph about HII in section 3.2:*

Sanderson et al. (2002) developed a human influence index (HII) dataset for mapping the areas with and without human footprint. The HII dataset quantifies human influence on terrestrial ecosystems based on four proxies (nine datasets), including human population pressure (population density), land transformation (land use/cover, roads and railways, built-up centers, settlements), accessibility (roads and railways, coastlines, navigable rivers) and electrical power infrastructure (night-time lights). Each of the nine datasets assigns a score from 0 to 10 according to a rating (or alternative in a single score and 0) to assess human influence on 1-km$^2$ land surface. Sum scores from nine datasets were standardized as HII values which range from 0 to 64 (WCS/CIESIN, 2005), with higher value indicating higher degree of human impact (Fig. 1b). HII reflects only modern human impact and doesn't consider the human impact during the past. The HII has been recently employed to assess human influence on pollen assemblages in China (Li et al., 2014; Li et al., 2015). In this study, HII values at 1-km$^2$ grids are assigned to modern pollen reference sites using ArcGIS.

- p. 4 part 3.3 see main point 4

*We have added the pollen diagram and a paragraph in the text (See above).*

- p.5, line 3: which statistical techniques have been tested? Could you write 1 sentence on the concept of the WAPLS?

*We have rewritten the sentences as: "The WA-PLS approach combines the virtues of WA method to model ecological optima of species and PLS method to select linear components from biological assemblages (ter Braak and Juggins, 1993). It has been tested, along with WA, MAT, and pollen response*

surface method (PRS), for eastern China data and demonstrated to give better results (Cao et al., 2014; Xu et al., 2010b) …"

- p.5, line 14: I don't understand, you also use MAT? Why? Please explain more and give more details on the methods

*We used MAT because relationship between pollen and HII is quite different to pollen and climate. HII is much more random and irregular comparing to climate variables, we can use it in ordination as an explanatory variable, but we cannot model HII optima or tolerance for a pollen taxon using WA or WA-PLS methods. There is no ecological basis to do like this. However, by measuring the mean analogue-HII, we have a chance to assess the potential bias induced from human impact on surface samples in reconstructing past climate. This is because the reconstructed climate of a fossil sample is mostly determined by those surface samples which have similar or close pollen spectra no matter what method is used.*

*We rewritten this part as:* "HII was also analysed in the same way to evaluate the human impact on the pollen data and the quality of the training set. HII as an environmental variable with certain features of stochastic on locations and intensity, there is no robust ecological basis to estimate the optima and tolerance for a pollen taxon using any pollen-HII calibration set. Therefore, we used an indirect method to assess the potential bias induced from training set due to human impact on surface samples. At first, we found five closest modern analogues for each fossil sample using MAT (Simpson, 2007), and then used mean HII value at the analogue sites to examine the human influence on the analogue samples and to evaluate the bias in climate reconstruction for that given fossil sample."

- p.5, line 21: "the differences between 151(147) and 99(93) taxa in the Natural (human) set are only explained by the exclusion of rare taxa?

*It can be explained like this. There is no need to omit the rare taxa from ecological perspective, and wo do this for reducing the "noise" which refers the component of variation arising from some random effects, such as pollen dispersal and deposition, analytical and counting errors, taxonomy discordance in data-sets merging, which all potentially increase the noise in a training set. With large-size and species-rich data, WA-PLS may fail to improve on WA due to complex noise (Juggins and Birks, 2012). Excluding some "rare data" is necessary in reconstructing work.*

- p.5, line 38: I don't understand why you don't keep the 3 component model (even given the threshold of 5%).

*We used threshold of 5% to select the 'minimal adequate model' to prevent over-fitting (Birks, 1998). The reduction in prediction error sometimes could to be caused by fitting improvement of only a few samples, and model with more components is not a guarantee to increase the predictive power, sometimes even poorer (Juggins and Birks, 2012). In our N-set, reduction in RMSEP using 3 component model is only 1.48%, and it is not significant (p=0.098) in randomisation t-test. Two-component is more appropriate in this case.*

*We also added the p-value (presented in Table 2) in the text:* "and not significantly different ($P = 0.098$)".

6. Discussion

- P.6, line 31 ….. is a function of climate: yes but also other factors play a role

*Agreed, we mentioned this in the next sentence: "…this indirect pollen-climate relationship can be affected by several other (non-climatic) factors". We have rewritten this sentence as "The pollen record is a complex and nonlinear function of vegetation which in turn is a function of climate based on some key assumptions".*

- p.7, line 18: a significant bias: do you mean statistically significant? it has been tested?

*We mean the reconstructed $P_{ann}$ values of early and middle Holocene using H-set are less than N-set and the difference is statistically significant (P<0.001).*

*We have rewritten it as:* Using the H-set in this study, significantly lower (P <0.001) $P_{ann}$ values relative to the N-set are reconstructed for Lake Gonghai during the early and middle Holocene (TW3 and TW4), when tree pollen contributed more than 40% to the total pollen sum. This suggests that drier biases may have also been induced from surface samples using the N-set for this period.

- p.8, line 16: how is calculated the "mean analogue-HII values in the N-set"?

*We used MAT method to find 5 best analogues in the both training sets for each fossil sample and calculated their mean HII values. We have added this information in the rewritten section 3.4.*

- p.8 line 32: "This raises the question of whether the Holocene climate could be quantitatively reconstructed using pollen data from eastern China. The answer is not a simple 'yes' or 'no'"; I think that the question is "when" and "where" given that your reconstruction before 2.8 ka can be considered as robust and climate driven

*This is indeed a very good and important comment. Farming practice in some places within study area was quite early, however, getting a sequence with less human disturbance before 2.8 ka is still possible, such as Gonghai. Indeed, the calibration set we called natural has been also effected by human impact from both the present and the past. The deviation between reconstructed $P_{ann}$ results is due to the human-induced bias on optima estimate in the H-set. We know that reference samples in the H-set are strongly affected by human activities. At least we can assume that the $P_{ann}$ reconstruction based on the N-set should be closer to the 'reality' in the past.*

7. Tables and figures

- Fig 1: a, b, the legend is not clear (the right one); the site is not easy to find on the fig: please correct

*Corrected. We have changed the font size in the legend box and the site symbol and its color (see below).*

**3.4 Numerical analyses**

Relationships between surface pollen spectra and climate variables are assessed by ordination techniques. To stabilise the variance and optimise the signal-to-noise ratio in the data, pollen taxa which occur in at least 3 samples and contribute $\geq 3\%$ in at least one sample were selected and square-root transformed for further analyses (Prentice, 1980). The length of the first axis in Detrended Correspondence Analysis (DCA; Hill and Gauch, 1980) was used to determine whether Redundancy Analysis (RDA) or Canonical Correspondence Analysis (CCA) should be chosen for the constrained ordination (ter Braak and Prentice, 1988). The ratio of the constrained eigenvalue to the first unconstrained eigenvalue ($\lambda_1/\lambda_2$) for a climate variable is used to assess its potential to be reconstructed (ter Braak, 1987). A value of $\lambda_1/\lambda_2$ greater than one suggests that the variable is the main determinant in the dataset; otherwise, the reconstruction of the variable should be conducted with caution (Juggins, 2013).

HII was also analysed in the same way to evaluate the human impact on the pollen data and the quality of the training set. HII as an environmental variable with certain features of stochastic on locations and intensity, there is no robust ecological basis to estimate the optima and tolerance for a pollen taxon using any pollen-HII calibration set. Therefore, we used an indirect method to assess the potential bias induced from training set due to human impact on surface samples. At first, we found five closest modern analogues for each fossil sample using MAT (Simpson, 2007), and then used mean HII value at the analogue sites to examine the human influence on the analogue samples and to evaluate the bias in climate reconstruction for that given fossil sample.

The WA-PLS approach combines the virtues of WA method to model ecological optima of species and PLS method to select linear components from biological assemblages (ter Braak and Juggins, 1993). It has been tested, along with WA, MAT, and pollen response surface method (PRS), for eastern China data and demonstrated to give better results (Cao et al., 2014; Xu et al., 2010b) due to its generally good performance under non-analogue situations and ability to cope with spatial autocorrelation (Cao et al., 2014; Juggins and Birks, 2012). The optimal number of WA-PLS components was selected using a randomisation t-test (van der Voet, 1994). Low root mean squared error of prediction (RMSEP), low average and maximum biases, a high coefficient of determination ($R^2$) between the predicted and observed climate values, and a rule-of-thumb threshold of 5% (reduction in RMSEP for adding a component) were all considered when selecting a model (Birks, 1998; Birks et al., 2010; Juggins and Birks, 2012).

[revised manuscript text omitted]

**5. Discussion**

**5.1 Climatic signals in pollen assemblages obscured by human impact**

The pollen record is a complex and nonlinear function of vegetation which in turn is a function of climate based on some key assumptions (Birks et al., 2010). The big challenge for pollen-based climate reconstructions is that this indirect pollen-climate relationship can be affected by several other (non-climatic) factors, for example, by human activities (Birks and Seppä, 2004; Ren, 2000; Xu et al., 2010b). RDA results show that the ability of $P_{ann}$ to explain pollen variance declines a lot in the H-set in comparison to the N-set (Table 1). The statistical performance of WA-PLS for the H-set is poorer (Table 2), suggesting climatic signals in the H-set have been partly obscured by human impact. Due to agricultural land use and human-induced deforestation of the plains and hilly areas (e.g. terraced fields) in the humid and sub-humid regions, tree pollen percentages decrease and herb percentages increase substantially in surface samples, even after excluding distinctly cultivated taxa. It is easy to imagine that herbaceous taxa, such as *Artemisia*, Chenopodiaceae, *Humulus* and weed-type *Poaceae* would expand after forest clearance and this will change the regional vegetation composition and relative pollen abundances (Ding et al., 2011; Li et al., 2015). It will also alter the pollen-climate relationships for many pollen taxa in the response models (St. Jacques et al., 2008). This alteration can be seen in the comparisons of RDA ordination (Fig. 3) and estimated WA optima and tolerances (Fig. 7) for 15 major taxa between two datasets.

Selected pollen taxa can be separated into two tree and herb groups by contrasting their WA optima in the N- and H-sets (Fig. 7). The inferred optima of most woody taxa in the H-set are shifted towards drier conditions and their tolerances compressed. This means that a fossil pollen assemblage with a high proportion of woody taxa would be assigned a lower $P_{ann}$ value when the H-set is employed in the transfer function. Conversely, $P_{ann}$ values will be overestimated when herbaceous taxa dominate in a fossil sample. This is clearly seen in the $P_{ann}$ curves for Lake Gonghai (Fig. 6a). The reconstructed $P_{ann}$ deviations between the N- and H-sets and the tree pollen percentage curve demonstrate similar trends (Fig. 6) and are statistically correlated (R = 0.81). However, the $P_{ann}$ deviation is not simply determined by the proportion of tree and herb taxa. For example, the deviations of (time-window) TW2 and the later TW5 are both around zero, but the tree pollen comprises 20–30% and 40–50%, respectively, rather than being equal with the percentage of herbs. This is a consequence of the vegetation composition and species characteristics.

In short, human impact obscures the climatic signals in pollen spectra by distorting the response relationship between pollen abundance and climate (Birks et al., 2010; Seppä et al., 2004), thus influencing the assumed climatic optima and tolerances of pollen taxa in the model (Fig. 7). When such a human-influenced calibration set is applied to a fossil record, which represents mostly natural vegetation, a more or less serious bias in the reconstructed past climate should be expected (St. Jacques et al., 2008; Xu et al., 2010a). Using the H-set in this study, significantly lower (P <0.001) $P_{ann}$ values relative to the N-set are reconstructed for Lake Gonghai during the early and middle Holocene (TW3 and TW4), when tree pollen contributed more than 40% to the total pollen sum. This suggests that drier biases may have also been induced from surface samples using the N-set for this period. 
[revised manuscript text omitted]

Bartlein, P. J., Harrison, S. P., Brewer, S., Connor, S., Davis, B. A. S., Gajewski, K., Guiot, J., Harrison-Prentice, T. I., Henderson, A., Peyron, O., Prentice, I. C., Scholze, M., Seppä, H., Shuman, B., Sugita, S., Thompson, R. S., Viau, A. E., Williams, J., and Wu, H.: Pollen-based continental climate reconstructions at 6 and 21 ka: a global synthesis, Clim. Dyn., 37, 775–802, 2010.

Bestel, S., Crawford, G. W., Liu, L., Shi, J., Song, Y., and Chen, X.: The evolution of millet domestication, Middle Yellow River Region, North China: Evidence from charred seeds at the late Upper Paleolithic Shizitan Locality 9 site, Holocene, 24, 261–265, 2014.

Birks, H. J. B.: Numerical tools in palaeolimnology - Progress, potentialities, and problems, J. Paleolimnol., 20, 307–332, 1998.

Birks, H. J. B., Heiri, O., Seppä, H., and Bjune, A. E.: Strengths and weaknesses of quantitative climate reconstructions based on Late-Quaternary biological proxies, The Open Ecology Journal, 3, 68–110, 2010.

Birks, H. J. B., Line, J. M., Juggins, S., Stevenson, A. C., and Terbraak, C. J. F.: Diatoms and pH Reconstruction, Philosophical Transactions of the Royal Society B-Biological Sciences, 327, 263–278, 1990.

Birks, H. J. B. and Seppä, H.: Pollen-based reconstructions of late-Qaternary climate in Europe–progress, problems, and pitfalls, Acta Palaeobot., 44, 317–334, 2004.

Cao, X., Herzschuh, U., Telford, R. J., and Ni, J.: A modern pollen–climate dataset from China and Mongolia: Assessing its potential for climate reconstruction, Rev. Palaeobot. Palynol., 211, 87–96, 2014.

Cao, X., Ni, J., Herzschuh, U., Wang, Y., and Zhao, Y.: A late Quaternary pollen dataset from eastern continental Asia for vegetation and climate reconstructions: Set up and evaluation, Rev. Palaeobot. Palynol., 194, 21–37, 2013.

Cao, X., Xu, Q., Jing, Z., Tang, J., Li, Y., and Tian, F.: Holocene climate change and human impacts implied from the pollen records in Anyang, central China, Quat. Int., 227, 3–9, 2010.

Chen, F., Xu, Q., Chen, J., Birks, H. J., Liu, J., Zhang, S., Jin, L., An, C., Telford, R. J., Cao, X., Wang, Z., Zhang, X., Selvaraj, K., Lu, H.,

Li, Y., Zheng, Z., Wang, H., Zhou, A., Dong, G., Zhang, J., Huang, X., Bloemendal, J., and Rao, Z.: East Asian summer monsoon precipitation variability since the last deglaciation, Sci. Rep., 5, 11186, 2015.

Crawford, G.: Early rice exploitation in the lower Yangzi valley: What are we missing?, Holocene, 22, 613–621, 2011.

Ding, W., Pang, R., Xu, Q., Li, Y., and Cao, X.: Surface pollen assemblages as indicators of human impact in the warm temperate hilly areas of eastern China, Chin. Sci. Bull., 56, 996–1004, 2011.

Domrös, M. and Peng, G.: The Climate of China, Springer, Berlin, 1988.

Editorial Committee of Vegetation Map of China, CAS: Vegetation map of the People's Republic of China (1:1000000), Geological Publishing House, Beijing, 2007.

Fu, C., Jiang, Z., Guan, Z., He, J., and Xu, Z. (Eds.): Regional climate studies of China, Springer, Berlin, 2008.

Grimm, E. C.: CONISS: a FORTRAN 77 program for stratigraphically constrained cluster analysis by the method of incremental sum of squares, Comput. Geosci., 13, 13–35, 1987.

Grimm, E. C.: Tilia 1.7. 16. In: Illinois State Museum, Research and Collection Center, Springfield, 2011.

[revised manuscript text omitted]

**Figure 5.** Proportion of variance (solid lines) in Gonghai Lake fossil pollen data explained by annual precipitation ($P_{ann}$) transfer functions with (a) the natural (N-) set and (b) the human-induced (H-) set. The thick black dotted lines indicate the proportion of variance explained by the first axis of a principal components analysis (PCA) and the fine red dotted lines indicate the 0.05 significance level. Histograms show the amount of variance explained by 999 transfer functions with random data.

[Figure]

**Figure 6.** Fossil pollen record of Lake Gonghai: (a) reconstructed annual precipitation using a natural (N-) set (blue) with upper side standard error of prediction (SEP) and a human-induced (H-) set (red) with lower side SEP. The modern instrumental value is marked on the scale axis for comparison (purple arrow). (b) Deviations in the reconstructed precipitation (grey) between the N-set and H-set based transfer functions with a 5-point moving average smoother (black). (c) The proportion of tree pollen taxa (%), and (d) the mean HII value of the five best analogues in the N-set (blue, with lower side standard deviation) and the H-set (red, with higher side standard deviation) for fossil samples. Six time windows (TWs), delineated according to the deviation pattern between the two reconstructions, are separated by grey dashed lines.

[Figure]

**Figure 7.** Caterpillar plot of weighted average (WA) optima and tolerances for 15 major pollen taxa in response to annual precipitation ($P_{ann}$). The taxa are arranged by optima values and taxa groups. Human impact generally shifts the inferred optima of woody taxa and herb taxa in opposite directions and compresses the tolerances for most taxa in the study area.